# Learning Robust Decision Policies
# from Observational Data

**Muhammad Osama**
muhammad.osama@it.uu.se

**Dave Zachariah**
dave.zachariah@it.uu.se

**Peter Stoica**
peter.stoica@it.uu.se

Division of System and Control
Department of Information Technology
Uppsala University, Sweden.

## Abstract

We address the problem of learning a decision policy from observational data of past decisions in contexts with features and associated outcomes. The past policy maybe unknown and in safety-critical applications, such as medical decision support, it is of interest to learn robust policies that reduce the risk of outcomes with high costs. In this paper, we develop a method for learning policies that reduce tails of the cost distribution at a specified level and, moreover, provide a statistically valid bound on the cost of each decision. These properties are valid under finite samples – even in scenarios with uneven or no overlap between features for different decisions in the observed data – by building on recent results in conformal prediction. The performance and statistical properties of the proposed method are illustrated using both real and synthetic data.

## 1   Introduction

We consider data of discrete decisions $x \in \mathcal{X}$ taken in contexts with features $\mathbf{z} \in \mathcal{Z}$. The outcome of each decision has an associated cost $y \in \mathcal{Y}$ (or equivalently, negative reward). For instance, we may obtain data from a hospital in which patients with features $\mathbf{z}$ are given treatment $x$ to lower their blood pressure and $y$ denotes the change of pressure value. The observational data is drawn independently as follows

$$(x_i, y_i, \mathbf{z}_i) \sim p(x, y, \mathbf{z}) = p(\mathbf{z}) \underbrace{p(x|\mathbf{z})}_{\text{past policy}} p(y|x, \mathbf{z}), \quad i = 1, \ldots, n \qquad (1)$$

where we have used a causal factorization of the unknown data-generating process. The distribution of contexts is described by $p(\mathbf{z})$ and $p(x|\mathbf{z})$ summarizes a decision policy which is generally unknown. We assume that there are no unobserved confounders.

Using the $n$ training data points, our goal is to automatically improve upon the past policy. That is, learn a new policy, which is a mapping from features to decisions

$$\pi(\mathbf{z}) : \mathcal{Z} \to \mathcal{X},$$

such that the outcome cost $y$ will tend to be lower than in the past. This policy partitions the feature space $\mathcal{Z}$ into $|\mathcal{X}|$ disjoint regions. A sample from the resulting data generating process can then be expressed as

$$(x, y, \mathbf{z}) \sim p^{\pi}(x, y, \mathbf{z}) = p(\mathbf{z})1\{x = \pi(\mathbf{z})\}p(y|x, \mathbf{z}), \qquad (2)$$

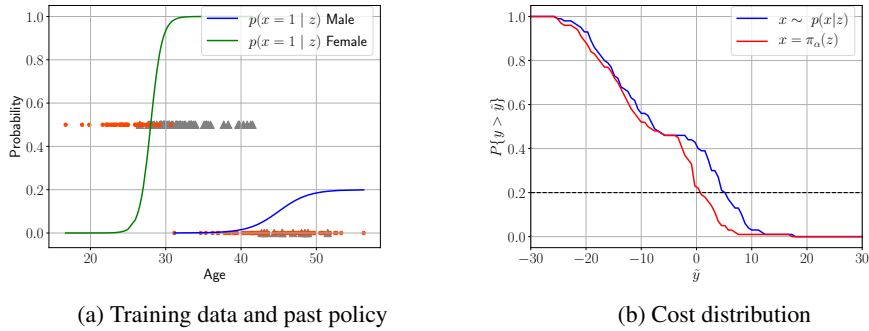

(a) Training data and past policy      (b) Cost distribution

Figure 1: Example of synthetic patient data with features $\mathbf{z} \in \{\texttt{age}, \texttt{gender}\}$ and decisions $x \in \{0, 1\}$ on whether or not to assign a treatment against high blood pressure. The outcome cost $y \in [-30, 30]$ here is the change in blood pressure. (a) Training data $(x_i, \mathbf{z}_i)$ where treatment and no treatment are denoted by $\triangle$ and $\bullet$, respectively. The example illustrates a past policy with highly uneven feature overlap, such that $p(x = 1|\mathbf{z})$ approaches 0 for younger males and 1 for older women, respectively. (b) Probability that a change of blood pressure $y$ exceeds a level $\widetilde{y}$, when $x$ is assigned according to past policy $p(x|\mathbf{z})$ vs. a proposed robust policy $\pi_\alpha(\mathbf{z})$ that targets lowering the tail costs at the $\alpha = 20\%$ level (dashed line).

where $1\{\cdot\}$ is the indicator function. In the treatment regime literature [17, 24, 25, 7, 21], the conventional aim is to minimize the expected cost, viz.

$$\min_{\pi \in \Pi} \mathbb{E}^\pi[y], \quad \text{where } \mathbb{E}^\pi[y] \equiv \mathbb{E}\left[\sum_{x \in \mathcal{X}} \mathbb{E}[y|x, \mathbf{z}]1\{x = \pi(\mathbf{z})\}\right] \equiv \sum_{x \in \mathcal{X}} \mathbb{E}\left[\frac{1\{x = \pi(\mathbf{z})\}}{p(x|\mathbf{z})}y\right], \quad (3)$$

where the last identity follows if features overlap across decisions so that $p(x|\mathbf{z}) > 0$ [9]. The optimal policy for this problem is $\pi(\mathbf{z}) = \arg\min_{x \in \mathcal{X}} \mathbb{E}[y|x, \mathbf{z}]$ and is determined by the unknown training distribution (1). Thus a policy must be learned from $n$ training samples, where a fundamental source of uncertainty about outcomes is uneven feature overlap across decisions [4, 11] (see Fig. 1a for an illustration). Eq. (3) is equivalent to an off-policy learning problem in contextual bandit settings using logged data [13, 6, 19, 10, 18], but where the past policy is unknown.

A common approach is to learn a regression model of $\mathbb{E}[y|x, \mathbf{z}]$, which in the case of binary decisions $\mathcal{X} = \{0, 1\}$ and linear models restricts the class of policies to the form $\Pi_\gamma = \{\pi(\mathbf{z}) = 1\{\boldsymbol{\gamma}^\top \mathbf{z} + \gamma_0 > 0\}\}$. To avoid the sensitivity to regression model misspecification, an alternative approach is to learn a model of $p(x|\mathbf{z})$ and then approximately solve (3) by numerical search over a restricted parametric class of policies $\Pi_\gamma$. In scenarios with highly uneven feature overlap, however, this approach leads to high-variance estimates of $\mathbb{E}^\pi[y]$, see the analysis in [6].

Reliably estimating the expected cost of a policy would yield an important performance certificate in safety-critical applications [21]. In such applications, however, reducing the prevalence of high-cost outcomes is a more robust strategy than reducing the expected cost, even when such tail events have low probability, see Figure 1b for an illustration. This is especially relevant when the conditional distribution of outcome costs $p(y|x, \mathbf{z})$ is skewed or has a dispersion that varies with $x$ [23].

In this paper, we develop a method for learning a robust policy that

- targets the reduction of the tail of the cost distribution $p^\pi(y)$, rather than $\mathbb{E}^\pi[y]$,

- provides a statistically valid limit $y_\alpha(\mathbf{z}) \geq y$ for each decision,

- is operational even when there is little feature overlap.

Moreover, when the past policy is unknown, the robust policy can be learned using unsupervised techniques, which obviates the need to specify associative models $\widehat{\mathbb{E}}[y|x, \mathbf{z}]$ and/or $\widehat{p}(x|\mathbf{z})$. The method is demonstrated using both real and synthetic data.

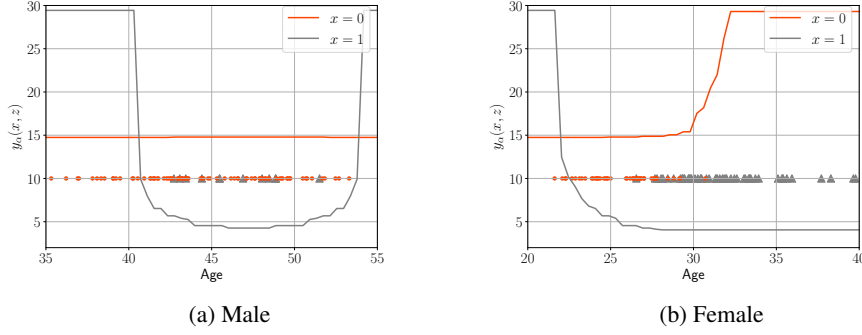

|            | (a) Male | (b) Female |
|---|---|---|

Figure 2: Synthetic patient data from Figure 1 along with limit $y_\alpha(x, \mathbf{z})$ such that $\mathbb{P}^x\{y \leq y_\alpha(x, \mathbf{z})\}$ is no lower than $1 - \alpha = 80\%$. The functions are learned using the method described in Sections 3.1 and 3.2. The training data provides evidence that treating (a) males in ages 41-53 years and (b) females in ages 22-40 years, yields the lowest tail costs. A robust policy $\pi_\alpha(\mathbf{z})$ in (5) selects decisions in $\mathcal{X}$ which yields the minimum $y_\alpha(x, \mathbf{z})$ and therefore targets the reduction of the tail of $p^\pi(y)$ (Fig. 1b). For younger males, however, data on treatment is unavailable and the limit becomes uninformative, $y_\alpha(x, \mathbf{z}) = \max(\mathcal{Y})$. Conversely, data on untreated older female is unavailable.

## 2   Problem formulation

We consider a policy $\pi(\mathbf{z})$ to be robust if it can reduce the tail costs at a specified level $\alpha$ as compared to the past policy – even for finite $n$ and highly uneven feature overlap. We define the $\alpha$-tail as all $y_\alpha$ for which the probability $\mathbb{P}^\pi\{y \leq y_\alpha\}$ is greater than or equal to $1 - \alpha$. An *optimal* robust policy therefore minimizes the $(1 - \alpha)$-quantile of the cost, viz. a solution to

$$\min_{\pi \in \Pi} \inf \left\{ y_\alpha \in \mathcal{Y} : \mathbb{P}^\pi\{y \leq y_\alpha\} \geq 1 - \alpha \right\}, \tag{4}$$

Since a learned policy $\pi(\mathbf{z}; \mathcal{D}_n)$ is a function of the training data $\mathcal{D}_n = \{(x_i, y_i, \mathbf{z}_i)\}_{i=1}^n$, the probability is also defined over all $n$ i.i.d. training points.

The problem we consider is to learn a policy in a class $\Pi_\alpha$ that approximately solves (4) and certifies each decision by a limit $y_\alpha(\mathbf{z}) \geq y$ that holds with a probability of at least $(1 - \alpha)$ for finite $n$ and highly uneven feature overlap.

## 3   Learning Method

Since the cumulative distribution function (CDF) in (4) is unknown for a given policy, it is a challenging task to find the minimum $y_\alpha$ which satisfies the $(1 - \alpha)$ constraint. We propose to restrict the policies to a class $\Pi_\alpha$, constructed as follows: Suppose there exists a feature-specific limit $y_\alpha(x, \mathbf{z})$ for a given decision $x \in \mathcal{X}$, such that $\mathbb{P}^x\{y \leq y_\alpha(x, \mathbf{z})\}$ is no less than $1 - \alpha$. Here $\mathbb{P}^x = \mathbb{P}^{\pi=x}$ is a short-hand notation when enforcing a decision $x$. Then we define $\Pi_\alpha$ as all policies $\pi(\mathbf{z})$ that select $x$ with the minimum cost limit at the specified level $\alpha$. That is, a class of robust policies

$$\Pi_\alpha = \left\{ \pi(\mathbf{z}) = \arg\min_{x \in \mathcal{X}} y_\alpha(x, \mathbf{z}) \; : \; \mathbb{P}^x\left\{y \leq y_\alpha(x, \mathbf{z})\right\} \geq 1 - \alpha, \; \forall x \in \mathcal{X} \right\} \tag{5}$$

Learning a policy in $\Pi_\alpha$ therefore amounts to using $\mathcal{D}_n$ to learn a set of functions $\{y_\alpha(x, \mathbf{z})\}_{x \in \mathcal{X}}$ that satisfy the constraints. Figure 2 illustrates $y_\alpha(x, \mathbf{z})$ constructed using the method described below, for a binary decision variable $x$.

*Remark:* If there is a tie among $\{y_\alpha(x, \mathbf{z})\}_{x \in \mathcal{X}}$, the policy can randomly draw $x$ from the minimizers. If the limits are non-informative, $y_\alpha(x, \mathbf{z}) \equiv \max(\mathcal{Y})$, the method will indicate that the data is not sufficiently informative for reliable cost-reducing decisions. See Figure 2 for regions in feature space where there is no data about outcomes for treated younger males and untreated older women; consequently $y_\alpha(x, \mathbf{z}) = \max(\mathcal{Y})$ for such pairs of features and decisions.

## 3.1 Statistically valid limits

To construct feature-specific limits $y_\alpha(x, \mathbf{z})$ that satisfy the constraint in (5), we leverage recent results developed using the conformal prediction framework [22, 14, 1]. We begin by quantifying the divergence of a sample $(x, y, \mathbf{z})$ in (2) from those in $\mathcal{D}_n$, using the residual

$$s(x, y, \mathbf{z}) = |y - \mu(x, y, \mathbf{z})| \geq 0, \tag{6}$$

where $\mu(x, y, \mathbf{z})$ is any predictor of the cost fitted using $\mathcal{D}_n \cup (x, y, \mathbf{z})$. Then $s(x, y, \mathbf{z})$ can be viewed as a random non-conformity score with a CDF $F(s)$ and quantile

$$s_{1-\alpha}(F) = \inf\{s : F(s) \geq 1 - \alpha\} \tag{7}$$

**Result 1** (Finite-sample validity). *For a given level $\alpha$ and context $\mathbf{z}$, construct a set of probability weights*

$$p_k(x_i, \mathbf{z}_i) \triangleq \frac{w_k(x_i, \mathbf{z}_i)}{\sum_{j=1}^n w_k(x_j, \mathbf{z}_j) + w_k(x, \mathbf{z})}, \quad \text{where } w_k(x, \mathbf{z}) \triangleq \frac{1\{x = k\}p(\mathbf{z})}{p(\mathbf{z}|x)p(x)}, \tag{8}$$

*for $k \in \mathcal{X}$ and define an empirical cdf for the residuals*

$$\widehat{F}_x(s) = \sum_{i=1}^n p_x(x_i, \mathbf{z}_i)1\{s \geq s_i\} + p_x(x, \mathbf{z})1\{s \geq s(x, y, \mathbf{z})\}, \tag{9}$$

*where $s_i = |y_i - \mu(x_i, y_i, \mathbf{z}_i)|$. Then*

$$y_\alpha(x, \mathbf{z}) \triangleq \max\left\{y \in \mathcal{Y} : s(x, y, \mathbf{z}) \leq s_{1-\alpha}(\widehat{F}_x)\right\}, \tag{10}$$

*satisfies the probabilistic constraint $\mathbb{P}^x\{y \leq y_\alpha(x, \mathbf{z})\} \geq 1 - \alpha$ in (5).*

*Proof.* By expressing $w_k(x, \mathbf{z}) \equiv \frac{q_k(x|\mathbf{z})p(\mathbf{z})}{p(x|\mathbf{z})p(\mathbf{z})}$, where $q_k(x|\mathbf{z}) = 1\{x = k\}$ it follows from [1, corr. 1] that the set in (10) will cover $y$ with a probability of at least $1 - \alpha$. $\qquad\square$

Computing $y_\alpha(x, \mathbf{z})$ requires a search of the maximum value in the set (10), which can be implemented efficiently using interval halving. Each evaluation point in the set, however, requires re-fitting $\mu(x, y, \mathbf{z})$ to $\mathcal{D}_n \cup (x, y, \mathbf{z})$ in (6). Therefore for an efficient computation of (10), we consider the locally weighted average of costs, i.e.,

$$\mu(x, y, \mathbf{z}) = \sum_{i=1}^n p_x(x_i, \mathbf{z}_i)y_i + p_x(x, \mathbf{z})y \tag{11}$$

which is linear in $y$. The non-parametric model in (11) also avoids yielding conformal intervals more sensitive to model misspecification in case of a parametric model for $\mu(x, y, \mathbf{z})$. This choice then defines a policy in $\Pi_\alpha$ and is illustrated in Figures 3a and 3b. Each decision of the policy can then be certified by a limit $y_\alpha(\mathbf{z}) \geq y$ obtained by setting $y_\alpha(\mathbf{z}) = y_\alpha(\pi(\mathbf{z}), \mathbf{z})$ in (10) and the probability of exceeding the limit is bounded by $\alpha$. For the sake of clarity, the computation of $y_\alpha(\mathbf{z})$ is summarized in Algorithm 1. The computation of (7) requires sorting and therefore bounds the runtime of the Algorithm by $\mathcal{O}(n \log n)$.

An important property of (10) is that it is statistically valid also for highly uneven feature overlap. As $p(\mathbf{z}|x)$ approaches 0 for a given $x$, the probability weights in (8) concentrate so that $p_x(x, \mathbf{z}) \to 1$ in (9). Consequently, $y_\alpha(x, \mathbf{z})$ converges to $\max(\mathcal{Y})$ so that the proposed robust policy avoids decisions $x$ in contexts $\mathbf{z}$ for which there is little or no training data.

*Remark:* The proposed method is readily extendable to other known feature distributions $q(\mathbf{z})$ than the unknown $p(\mathbf{z})$ from which training data was obtained. This will only effect the evaluation of weights with $p(\mathbf{z})$ in the numerator of (8) replaced with $q(\mathbf{z})$.

**Algorithm 1** Robust policy

---

1: **Input**: Training data $\mathcal{D}_n$, level $\alpha$ and feature $\mathbf{z}$
2: **for** $x \in \mathcal{X}$ **do**
3:     Compute $\{p_x(x_i, \mathbf{z}_i)\}_{i=1}^n$ in (8)
4:     Set $\mu_0 = \sum_{i=1}^n p_x(x_i, \mathbf{z}_i)y_i$
5:     Set $\mathcal{Y}_\alpha := \emptyset$
6:     **for** $y \in \mathcal{Y}$ **do**
7:         Predictor $\mu(x, y, \mathbf{z}) := \mu_0 + p_x(x, \mathbf{z})y$
8:         Score $s(x, y, \mathbf{z}) := |y - \mu(x, y, \mathbf{z})|$
9:         CDF $\widehat{F}_x(s)$ in (9)
10:         **if** $s(x, y, \mathbf{z}) \leq s_{1-\alpha}(\widehat{F}_x)$ **then**
11:             $\mathcal{Y}_\alpha := \mathcal{Y}_\alpha \cup \{y\}$
12:         **end if**
13:     **end for**
14:     $y_\alpha(x, \mathbf{z}) = \max(\mathcal{Y}_\alpha)$
15: **end for**
16: **Output**: $\pi(\mathbf{z}) = \arg\min_x y_\alpha(x, \mathbf{z})$ and $y_\alpha(\mathbf{z}) = y_\alpha(\pi(\mathbf{z}), \mathbf{z})$

---

## 3.2 Unsupervised learning of weights

In randomized control trials, and other controlled experiments, the weights in (8) are given by a known past policy. In the general case, however, $w_k(x, \mathbf{z})$ must be learned from training data. This is effectively an unsupervised learning problem which therefore circuments the need for specifying associative models of $\mathbb{E}[y|x, \mathbf{z}]$ (regression) or $p(x|\mathbf{z})$ (propensity score).

The categorical distribution of past decisions, $p(x)$, is readily modeled as $\widehat{p}(x = k) = n^{-1} \sum_{i=1}^n 1\{x_i = k\}$ using $\mathcal{D}_n$. The conditional feature distribution $p(\mathbf{z}|x = k)$ can in turn be modelled by a flexible generative model, e.g. Gaussian mixture models or multinoulli models. The accuracy of the learned generative model $\widehat{p}(\mathbf{z}|x = k)$ can then be assessed using model validation methods, e.g. [15]. If the training data contains high-dimensional covariates, we propose constructing features $\mathbf{z}$ using dimension-reduction methods, such as autoencoders [2, 12, 16, 20]. The weights in (8) are learned via $\widehat{p}(x)$ and $\widehat{p}(\mathbf{z}|x)$, and using $\widehat{p}(\mathbf{z}) = \sum_{x \in \mathcal{X}} \widehat{p}(\mathbf{z}|x)\widehat{p}(x)$.

*Remark:* If a validated propensity score model already exists, one can simply use the equivalent form $w_k(x, \mathbf{z}) = 1\{x = k\}/\widehat{p}(x|\mathbf{z})$.

## 4 Numerical experiments

We study the statistical properties of policies in the robust class $\Pi_\alpha$, which we denote $\pi_\alpha(\mathbf{z})$. To illustrate some key differences between a mean-optimal policy (3) and a robust policy, we first consider a well-specified scenario in which the mean-optimal policy belongs to a given class $\Pi_\gamma$. Subsequently, we study a scenario with misspecified models using real training data. The code for the experiments can be found here.

## 4.1 Synthetic data

We consider a scenario in which patients are assigned treatments to reduce their blood pressure. We create a synthetic dataset, drawing $n = 200$ data points from the training distribution (1) where features $\mathbf{z} = [z_1, z_2]^\top$ represent age $z_1 \in \mathbb{R}$ and gender $z_2 \in \{0, 1\}$ (1 for females and 0 for males). The feature distribution for the population of patients $p(\mathbf{z})$ is specified as

$$p(z_1|z_2) = z_2 \times \mathcal{N}(30, 5) + (1 - z_2) \times \mathcal{N}(45, 5) \quad \text{and} \quad p(z_2) \equiv 0.5 \qquad (12)$$

The treatment decision $x \in \{0, 1\}$ is assigned based on a past policy which we specify by the probability

$$p(x = 1|\mathbf{z}) = z_2 \times 0.92\, f\left(-\frac{z_1 - 20}{6}\right) + (1 - z_2) \times 0.20\, f\left(-\frac{z_1 - 45}{2}\right), \qquad (13)$$

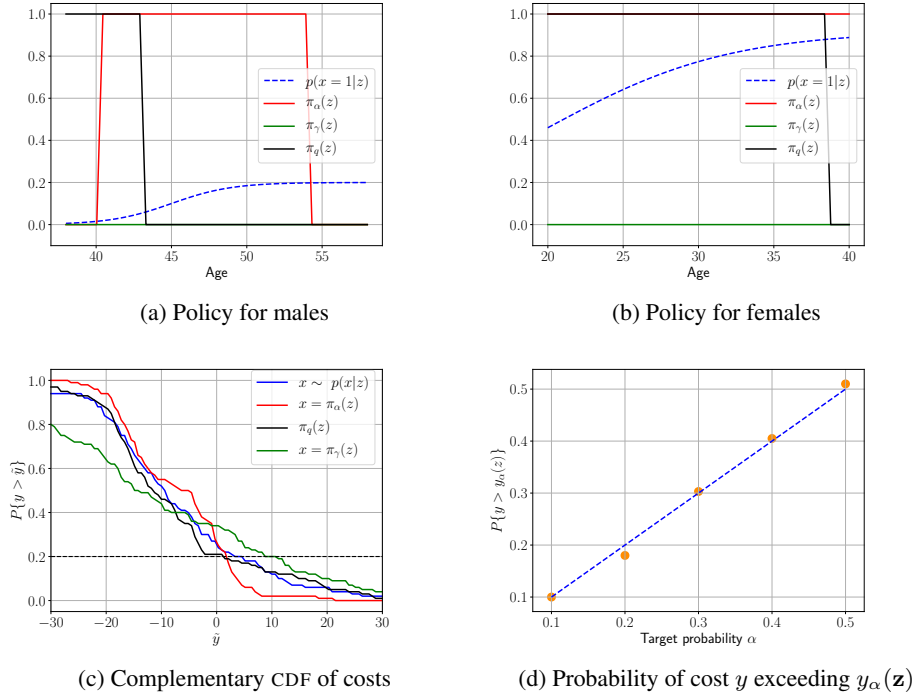

| (a) Policy for males | (b) Policy for females |
|---|---|
| (c) Complementary CDF of costs | (d) Probability of cost $y$ exceeding $y_\alpha(\mathbf{z})$ |

Figure 3: (a) and (b) show a past policy $p(x|\mathbf{z})$ with uneven feature overlap. Robust policy $\pi_\alpha(\mathbf{z})$, mean-optimal and quantile-optimal linear policies $\pi_\gamma(\mathbf{z})$ and $\pi_q(\mathbf{z})$ are learned from $\mathcal{D}_n$. The robust policy targets reducing tail costs at the $\alpha = 20\%$ level. Note that the mean-optimal policy is to not treat, i.e., $x = 0$. (c) Costs of past and learned policies along with $\alpha$-level (dashed). (d) Accuracy of the limit $y_\alpha(\mathbf{z})$ vs. different $\alpha$-levels, using 300 Monte Carlo runs.

where $f(a) = 1/(1 + \exp(-a))$ is the sigmoid function. See Figures 3a and 3b for all illustration. While the assignment mechanism is not necessarily realistic, we use it to illustrate the relevant case of uneven feature overlap. Finally, the change in blood pressure $y$ is drawn randomly as

$$p(y|x, \mathbf{z}) = x \times \mathcal{N}\left(\mathbf{e}_1^\top \mathbf{z} - 45,\ \sigma_1^2\right) + (1 - x) \times \mathcal{N}\left(\mathbf{e}_1^\top \mathbf{z} - 46,\ \sigma_0^2\right), \tag{14}$$

where $\sigma_1 = 0.2$ and $\sigma_0 = 20$. While the expected cost for the untreated group is lower than for the treated group, we consider the untreated patients to have more heterogeneous outcomes, so that the dispersion is higher. That is, $\mathbb{E}[y|0, \mathbf{z}] < \mathbb{E}[y|1, \mathbf{z}]$ while $\sigma_0 > \sigma_1$.

Since the past policy is unknown, we learn weights (8) for $\pi_\alpha(\mathbf{z})$ in an unsupervised manner, using $\widehat{p}(\mathbf{z}|x) = \widehat{p}(z_1|z_2, x)\widehat{p}(z_2|x)$, where $\widehat{p}(z_1|z_2, x)$ is a misspecified Gaussian model and $\widehat{p}(z_2|x)$ is a Bernoulli model. We let $\alpha = 20\%$. As a baseline comparison, we consider minimizing the expected cost (3) for a linear policy class $\Pi_\gamma$. Since $\mathbb{E}[y|x, \mathbf{z}]$ is a linear function in $\mathbf{z}$, this is a well-specified scenario in which the mean-optimal policy belongs to $\Pi_\gamma$. We fit a correct linear model of the conditional mean and denote the resulting policy by $\pi_\gamma(\mathbf{z})$. We also compare against the quantile optimal policy $\pi_q(\mathbf{z}) \in \Pi_\gamma$ [23], which uses a consistent estimate of the $\alpha$-quantile of the cost.

Figures 3a and 3b show the decision $x$ taken by the robust and mean-optimal policy, $\pi_\alpha(\mathbf{z})$ and $\pi_\gamma(\mathbf{z})$, respectively, as a function of features $\mathbf{z}$. Note that (14) leads to a mean-optimal policy $\pi_\gamma(\mathbf{z}) \equiv 0$, since the expected cost for the untreated group is lower than that of the treated group. By contrast, the robust policy $\pi_\alpha(\mathbf{z})$ takes into account that the dispersion of costs is much higher for untreated patients and therefore assigns $x = 1$ to male patients in the age span 41-54 years as well as all females in the observable age span. To reduce the risk of increased blood pressure at the specified level, it therefore opts for treatments more often. This is highlighted in Figure 3c which shows the cost distribution, using the complementary CDF $\mathbb{P}^\pi\{y > \widetilde{y}\}$, for the different policies. We see that the robust policy safeguards against large increases in blood pressure, where the $(1 - \alpha)$ quantile is smaller than that for the mean-optimal policy. The quantile optimal policy yields a marginally a smaller $(1 - \alpha)$ quantile than the proposed robust policy but also notably higher tail costs.

An important feature of the proposed methodology is that each decision of the policy $\pi_\alpha(\mathbf{z})$ has an associated limit $y_\alpha(\mathbf{z})$, such that the probability of exceeding it, $\mathbb{P}^\pi\{y > y_\alpha(\mathbf{z})\}$, is bounded by $\alpha$. Figure 3d shows the estimated probability under the robust policy versus the target level $\alpha$. Despite the misspecification of the Gaussian model $\widehat{p}(z_1|z_2, x)$, the target $\alpha$ provides an accurate limit for the actual probability.

## 4.2 Infant Health and Development Program data

Next, the properties of the proposed method are studied using real data. We use data from the Infant Health and Development program (IHDP) [3], which investigated the effect of personalized home visits and intensive high-quality child care on the health of low birth-weight and premature infants [8]. The data for each child included a 25-dimensional covariate vector $\widetilde{\mathbf{z}}$, containing information on birth weight, head circumference, gender etc., standardized to zero mean and unit standard deviation, as well as a decision $x \in \{0, 1\}$ indicating whether a child received special medical care or not. The outcome cost $y$ is a child's cognitive underdevelopment score (simply a sign change of a development score).

The covariate distribution $p(\widetilde{\mathbf{z}})$ is unknown. The past policy, which we also treat as unknown, was in fact a randomized control experiment, so that $p(x = 1|\widetilde{\mathbf{z}})$ was a constant. This policy was found to be successful in improving cognitive scores of the treated children as compared to those in the control group. To obtain outcome costs for either decision in $\mathcal{X}$, we generate $y$ synthetically by the nonlinear associative models following [8, 5]:

$$y|x = 0, \widetilde{\mathbf{z}} \sim \mathcal{N}(-\exp[(\widetilde{\mathbf{z}} + 0.5\mathbf{1})^\top \boldsymbol{\beta}], \ \sigma_0) \quad \text{and} \quad y|x = 1, \widetilde{\mathbf{z}} \sim \mathcal{N}(-\widetilde{\mathbf{z}}^\top \boldsymbol{\beta} - \omega, \ \sigma_1), \qquad (15)$$

where we consider different dispersions below. Here $\omega$ is selected as described in [5] and [8] so that the effect of treatment on the treated is $1.5$. The unknown parameter $\boldsymbol{\beta}$ is a 25-dimensional vector of coefficients drawn randomly from $\{0, \ 0.1, \ 0.2, \ 0.3, \ 0.4\}$ with probabilities $\{0.6, 0.1, 0.1, 0.1, 0.1\}$, respectively, as specified in [8]. The IHDP data contains 747 data points and we randomly select a subset of $n = 600$ training points that form $\mathcal{D}_n$. The remaining $147$ points are used to evaluate learned policies.

To learn the weights (8) for the robust policy, we first reduce the 25-dimensional covariates $\widetilde{\mathbf{z}}$ into 4-dimensional features $\mathbf{z} = \texttt{enc}(\widetilde{\mathbf{z}})$ using an autoencoder [2, sec.7.1]. Then $\widehat{p}(\mathbf{z}|x)$ is a learned Gaussian mixture model with four mixture components and $\widehat{p}(x)$ is a learned Bernoulli model. Together the models define (8) and a robust policy $\pi_\alpha(\mathbf{z})$ is learned for the target probability $\alpha = 20\%$. For comparison, we also consider a linear policy $\pi_\gamma(\widetilde{\mathbf{z}})$ that aims to minimize the expected cost (3) using linear models of the conditional means. Note that such models are well-specified and misspecified for the treated and untreated outcomes in (15), respectively. In addition, we also compare against the quantile-optimal linear policy $\pi_q(\mathbf{z})$ [23].

Figure 4 shows the cost distribution for the past and learned policies when the dispersions in (15) are equal or different. We see that in the cases of equal dispersion in Figure 4a and higher dispersion for untreated in Figure 4c, the robust and optimal linear policies reduce the $(1 - \alpha)$-quantile of the cost $y_\alpha$ as compared to that for the past policy, where the robust policy does slightly better.

Since the treated group tends to have a lower mean cost than the untreated group in the training data, the linear policy tends to assign $x = 1$ to most patients in the test data. Moreover, the misspecified linear model leads to biased estimates of the expected cost and the resulting policy $\pi_\gamma(\widetilde{\mathbf{z}})$ cannot fully capture the non-linear partition of the feature space implied by the mean-optimal policy based on $\mathbb{E}[y|x, \widetilde{\mathbf{z}}]$.

Figure 4e shows the cost distribution when the treatment outcome costs have higher dispersion. The tendency toward treatment assignment by the linear policies results in higher tail costs. By contrast, the robust policy adapts to a higher cost dispersion in the treated group and assigns fewer treatments which results in resulting in smaller tail costs. In this case, the tail cost is more similar to the past policy since its proportion of (random) treatment assignments is small in the data.

The robust methodology also provides a certificate $y_\alpha(\mathbf{z})$ for each decision, as illustrated in Figures 4b, 4f and 4d with respect to two standardized covariates for each child in the test set. The probability that the cost $y$ exceeds $y_\alpha(\mathbf{z})$ is $18.6\%$, estimated using 500 Monte Carlo runs, which is close to and no greater than the targeted probability $\alpha = 20\%$ despite the model misspecification of $\widehat{p}(\mathbf{z}|x)$.

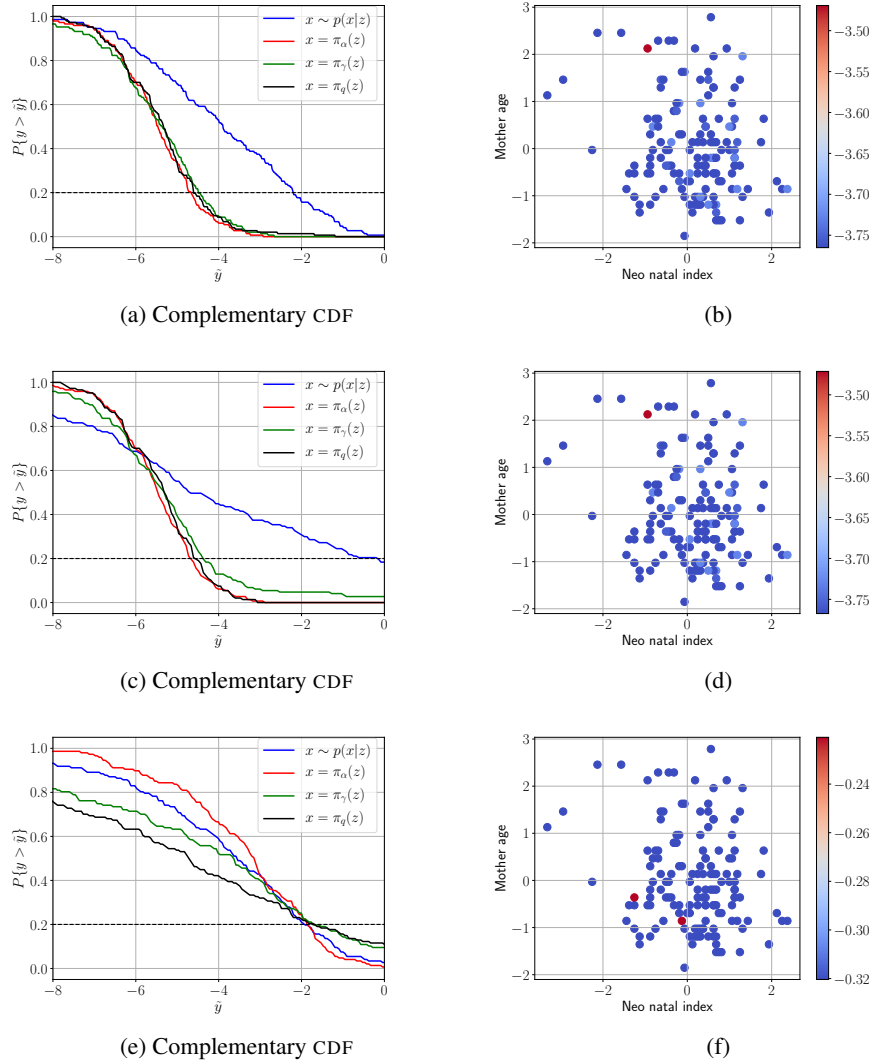

Figure 4: IHDP data and cognitive underdevelopment scores $y$. First column: Complementary CDFs for learned robust $\pi_\alpha$ and linear policies, respectively, as compared to past policy. We consider three different scenarios in (15): (a) $\sigma_1 = \sigma_0 = 1$, (b) $\sigma_1 = 1$, $\sigma_0 = 5$, and (c) $\sigma_1 = 5$, $\sigma_0 = 1$. Second column: limit $y_\alpha(\mathbf{z})$ (color bar) provided by the robust methodology, plotted against two standardized covariates: neonatal health index and mother's age of each child in the test data. Each unit corresponds to a standard deviation from the mean.

# 5 Conclusion

We have developed a method for learning decision policies from observational data that lower the tail costs of decisions at a specified level. This is relevant in safely-critical applications. By building on recent results in conformal prediction, the method also provides statistically valid bound on the cost of each decision. These properties are valid under finite samples and even in scenarios with highly uneven overlap between features for different decisions in the observed data. Using both real and synthetic data, we illustrated the statistical properties and performance of the proposed method.

## Broader Impact

We believe the work presented herein can provide a useful tool for decision support, especially in safety-critical applications where it is of interest to reduce the risk of incurring high costs. The methodology can leverage large and heterogeneous data on past decisions, contexts and outcomes, to improve human decision making, while providing an interpretable statistical guarantee for its recommendations. It is important, however, to consider the population from which the training data is obtained and used. If the method is deployed in a setting with a different population it may indeed fail to provide cost-reducing decisions. Moreover, if there are categories of features that are sensitive and subject to unwarranted biases, the population may need to be split into appropriate subpopulations or else the biases can be reproduced in the learned policies.

## Acknowledgments and Disclosure of Funding

This research was partially supported by the Swedish Research Council (contract no.: 2018-05040) and the *Wallenberg AI, Autonomous Systems and Software Program* (WASP) funded by Knut and Alice Wallenberg Foundation.

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
