[Reviews · NeurIPS 2020]

Review 1

Summary and Contributions: This manuscript explores the learning of policies from observational data that minimized an alternative cost function. Unlike most work that minimizes the expected cost, the authors focus on the tail of the cost at for every context. This tail is estimated using a result from conformal prediction. Finally, a demonstration of the algorithm is presented on simulations and real data.

Strengths: + Learning from observational data is an important area and looking at the tails is definitely a novel and interesting direction, relevant to the community. + Exploiting the result from conformal prediction is novel and sound. + The problem is clearly presented.

Weaknesses: There are many algorithmic points that I did not find addressed in this submission and would give supplemental insight, in addition to the application of the conformal prediction result. I appreciated the guarantee that the confidence interval is valid. However, is there any room to discuss optimality (an equivalent of power for hypothesis testing?) ? When would such a method break and produce a useless interval? What is the price (in terms of “power”) of using as predictor the locally weighted average of cost in Eq. (11)? Also, this predictor could be better motivated. Would it be useful to add some derivations in Appendix? Having a look at Algorithm 1, it is not clear to me what is the exact time complexity of this algorithm. This should be clearly discussed. Even with the approximation in (11), I am not sure whether that would scale to large datasets. Section 3.2 just explains how to use density estimation to form the probabilities used in (8). One natural question (treated in doubly robust estimation in its own way) is how is this confidence interval estimation robust to errors in the estimation of the propensity weights. The experiments have the merit of showing that the robust policy acts differently than the mean-optimal policy AND that the proposed algorithm controls the tail adequately. However, I think that more effort must be done to add reasonable baselines. For example, a generative model that would ignore the logging policy and model the tail seems minimal. And then, some similar model trained with importance sampling to correct for the logging? The linear policy is definitely too weak. Relationship to previous work must be discussed, there should be at least some similar discussions in the reinforcement learning community that must be brought up here. I have found one instance here [1]. That example is very applied, but it shows there *is* some related work, trying to estimate the noise in the reward distribution and to exploit the uncertainty in some way. [1] https://arxiv.org/pdf/1810.08700.pdf

Correctness: To my knowledge, what is presented in the paper seems reasonable.

Clarity: Some notations in the manuscript may be improved, P^pi and P^x are not defined. This is especially confusing at the beginning of the paper.

Relation to Prior Work: See my comment earlier, the paper could be better contextualized.

Reproducibility: Yes

Additional Feedback: After author feedback ----- I appreciate that the authors answered my questions and added supplementary baselines. I maintain my score and recommend acceptance of this manuscript.


Review 2

Summary and Contributions: This paper proposes a way to use conformal inference ideas to learn the optimal policy to maximize the reward at a chosen quantile. Doing so can improve robustness of policy learning to outlier rewards.

Strengths: The paper provides a strong technical presentation of policy learning and a certain notion of robustness to outlier observations. The proposed method delivers on providing guarantees with respect to this notion of robustness, and the paper presents a good evaluation on simulation and real data.

Weaknesses: The mathematical results seem to be a bit heuristic. In this setting, one would expect a requirement that p(x|z) >= \epsilon > 0, or something of the sort. That is, there must be "overlap" between the features z of people assigned policy x=0 and the features z of people assigned policy x=1. This never seems to be explicitly assumed. Similarly, there is no discussion of unobserved confounders. Importantly, these results all depend on a notion of counterfactuals, because the policy is being changed. If the policy in the observed data depends on some variable u that also affects y, but is unobserved, then when we deploy a policy that only depends on z, the performance may not be as we expect. Edit: Thanks to the authors for addressing these concerns, and pointing out the discussion of overlap for the main results. I still don't quite understand what happens when the estimate of p(x | z) = 0. I suppose perhaps in this case, one just chooses the best among the observed actions? I would appreciate more precision in describing the algorithm, and how these potentially infinite values are handled. I would also be interested to see if there was proof that the proposed method chooses a policy with better empirical overlap in the "Infant Health and Development Program" than baseline methods that don't optimize the tail costs. This would make the method very compelling and practical.

Correctness: The claims are all true, though implicitly depend on the unconfoundedness assumption. Perhaps the intention of saying that the distribution p(y, x, z) factorizes as p(z) p(x | z) p(y | x, z) was to implicitly say this, but it doesn't quite achieve that. For example, p(y | x, z) is well-defined even if there is a variable u correlated (and causal) for y and x, but in this case would not be what we need to know when deploying a new policy \pi(z) (that does not depend on u). I see two possible solutions: (1) is less mathematically precise, but more familiar to RL audiences, which is to say in words that you are assuming unconfoundedness, and therefore you assume that p(y | x, z) is generative. (2) is to be explicit, using potential outcomes notation to write y(x) as the reward that would've happened if action x was taken, and then use explicitly the unconfoundedness assumption to imply that p(y(x) | x, z) = p(y_observed | x, z). For references on this, see [1] Murphy, Susan A. "Optimal dynamic treatment regimes." Journal of the Royal Statistical Society: Series B (Statistical Methodology) 65.2 (2003): 331-355. [2] Namkoong, Hongseok, et al. "Off-policy Policy Evaluation For Sequential Decisions Under Unobserved Confounding." arXiv preprint arXiv:2003.05623 (2020). Murphy (2003) is a classical reference on this topic, but Namkoong et al. (2020) translates this into contemporary RL notation and has a discussion of why the unconfoundedness assumption is necessary in observational RL.

Clarity: Yes

Relation to Prior Work: Yes

Reproducibility: Yes

Additional Feedback: Updates: With the author feedback, my more careful reading of the discussion on overlap, and indication from the authors that they will try to be more explicit in their use of assumptions, I will increase my overall score for the submission. I hope the authors take my responses in the "weaknesses" section into consideration, and I think this has the potential to be an impactful paper.


Review 3

Summary and Contributions: The paper presents an approach for learning policies aimed at reducing the tails of the cost distribution. It also provides a statistically valid bound on the cost of each decision, which is an essential requirement in safety-critical decision-making areas like healthcare. Results on synthetic and real-world data demonstrate that the approach can learn robust policies even under feature overlap between decisions.

Strengths: 1. The paper is well-written and easy to follow. 2. The proposed approach for learning a robust policy is able to reduce the tail of the cost distribution rather than E[y] under the policy and provides a statistical bound on the cost of each decision. 3. The approach is valid under feature overlap as well as supported in Figure 2.

Weaknesses: 1. The divergence between a sample and the data is calculated by considering the residual. However, this approach may fail when the sample is shifted from the data distribution. In such settings, is there a way to adopt the method to account for the shift as well? 2. Since dimensionality reduction is performed in the real world experimental setting, it is not clear how the approach depends on the number of covariates. Is the proposed method better than a mean conditional policy for high-dimensional data as well? Adding some discussion on this can help situate the work in critical decision-making applications like healthcare, where we often find high-dimensional data. 3. What are the assumptions about the data-generating process? Are the statistical claims valid under confounding as well? 4. Though not the focus of the paper, it would be helpful to have some discussion around the fairness of the decisions and how $\alpha$ relates to unfairness?

Correctness: The experimental results are reasonable to establish the benefits of the proposed approach.

Clarity: The paper is well written as easy to follow.

Relation to Prior Work: I'd like to suggest some specifics for the related work as it seems incomplete, making it difficult to weigh the merits. Discussion concerning the following points will be helpful. 1. Observational data is often biased and it is crucial to discuss the relation specifically with respect to selection bias [1]. 2. Another issue is with respect to confounding. There are recent approaches in policy learning that also account for unobserved confounding [2]. 3. Since the ultimate goal is to make the decision based on the policy learned, it is especially important to consider whether the decisions are unfair [3]. [1] Atan, Onur, William R. Zame, and Mihaela van der Schaar. "Learning optimal policies from observational data." arXiv preprint arXiv:1802.08679 (2018). [2] Kallus, Nathan, and Angela Zhou. "Confounding-robust policy improvement." Advances in neural information processing systems. 2018. [3] Nabi, Razieh, Daniel Malinsky, and Ilya Shpitser. "Learning optimal fair policies." Proceedings of machine learning research 97 (2019): 4674.

Reproducibility: No

Additional Feedback: Updates upon feedback: After reading the author's response and other reviews my evaluation remains unchanged. This is interesting work which would strongly benefit from more focus on the relevant baselines and optimality. In particular, more discussion of these (including robustness to propensity weights, demerits of the linear policy) would lead to a stronger version of this paper. --------------------------------------------------------------------------------------- I have a few suggestions that can be taken into account. Some of them are mentioned above as well. 1. Having a simple example without feature overlap can help motivate the approach, and then following with example explained in Figure 1. 2. While the problem is exciting, and it's necessary to assess solutions for the same, a clear motivation for the proposed approach is missing.


Review 4

Summary and Contributions: This paper develops a robust method for learning policies

Strengths: The study is well-grounded in theory. It provides background on the method and has a real-world example.

Weaknesses: I think the greatest weakness is that the authors do not provide some form of comparison with other methods in their applications, or how they distinguish from previous methods. As far as I understand the applications "prove" the statistical validity but nothing more. This makes it difficult for me to judge the broader impact of this method.

Correctness: I was unable to verify all formula and proofs, but the terminology seems to be consistent.

Clarity: The paper is well written, there are only some typos (e.g. line 3 "may be").

Relation to Prior Work: There is no comparison to previous methods which makes it difficult .

Reproducibility: No

Additional Feedback:

[Author Response · NeurIPS 2020]

We thank the reviewers for their constructive feedback on our paper. We will take this feedback into account when revising the paper. Below we provide point-by-point response.

**Rev.#1**:

Reg. useless intervals and method breakdown: When $y_\alpha(x, \mathbf{z}) = \max(\mathcal{Y})$, the upper limit is noninformative which occurs when $(x, \mathbf{z})$ is 'far' from the training data. In such cases, however, the proposed policy would fall back onto the logging policy if $p(\mathbf{z}|x)$ is modelled accurately and used in the weights $w(x, \mathbf{z})$ in (8). The method would indeed perform poorly if the model were highly inaccurate, which would result in weights that may alter the decisions drastically. Fortunately, it is possible to use model validation techniques to assess the accuracy of the model, as pointed out in Sec. 3.2. Even in the case of misspecified models, the weights can be sufficiently accurate to provide accurate intervals, as Figure 3d demonstrates for the synthetic example in Sec. 4.1, where the empirical estimate of $\mathbb{P}^{\pi_\alpha}(y \leq y_\alpha(\mathbf{z}))$ matches the theoretical limit well.

Reg. motivation for locally weighted average $\mu(x, y, \mathbf{z})$: Firstly, it leads to computationally efficient conformal limits since $\mu(x, y, \mathbf{z})$ must be fitted for each evaluation at $(x, y, \mathbf{z})$. Using the nonparametric weighted average, each fitting has a constant runtime $\mathcal{O}(1)$. Secondly, using parametric predictive models $\breve{\mu}(x, y, \mathbf{z})$ yields conformal limits that are more sensitive to model misspecification. Indeed, misspecified parametric models may produce larger residuals and hence larger conformal limits than a nonparametric locally weighted average. These points will be highlighted in the revised manuscript.

Reg. complexity of Algorithm 1: The main operations which depend on the number of datapoints $n$ are lines 3, 4 and 10. Out of these, line 10 involves a sorting operation to compute the quantile $s_{1-\alpha}(\widehat{F})$ which dominates all other computations and results in a total runtime $\mathcal{O}(n \log n)$. The method thus is scalable to large $n$.

Reg. comparison to reasonable baselines: We have now added another baseline which explicitly learns a policy using an consistent estimate of $\alpha$-quantile of the costs $y$ (based on the cited paper by Wang et. al.). For the synthetic case in Sec. 4.1, it results in a slightly lower $\alpha$-quantile level, but significantly higher tail costs beyond the $\alpha$-quantile as compared to the proposed method. We will include comparisons with this additional baseline for both numerical examples in the revised manuscript or supplementary material.

Reg. discussion of relevant work in reinforcement learning: We will add the suggested references and additional references on safety-critical applications.

We will add a clarification on the notation $\mathbb{P}^x$.

**Rev.#2**:

Reg. unconfoundedness: We agree with the reviewer and we do assume that there are no unobserved confounders. We will make this assumption explicit in the revised manuscript.

Reg. overlap: Result 1 does require overlap $p(x|\mathbf{z}) > 0$ in order for the weights $w(x, \mathbf{z})$ in eq. (8) to be finite. However, as pointed out at the end of Sec. 3.1, as the evaluated weight $w(x, \mathbf{z}) \to \infty$, then $p_x(x, \mathbf{z}) \to 1$ and the conformal limit simply becomes uninformative so that the method remains operational even for infinite weight $w(x, \mathbf{z})$. We will clarify this point in the revised paper.

**Rev.#3**:

Reg. distributional shift: If the feature training distribution $p(\mathbf{z})$ shifts from the test distribution, say, $q(\mathbf{z})$, then the method can be readily extended to compensate for such distributional shifts, provided that $q(\mathbf{z})$ can be evaluated at a given $\mathbf{z}$. We will include this remark in the revised manuscript.

Reg. dimension reduction and high-dimensional data: We have not studied the effect of dimension reduction on the performance of our method. However, it is possible to check the accuracy of the learned generative model $\widehat{p}(\mathbf{z}|x = k)$ using the model validation methods referred to in Sec. 4.2. This provides a guideline for choosing the appropriate feature dimension to which data is can be reduced.

Reg. validity of results under confounding: Indeed, we do assume no unobserved confounders and will make the assumption explicit.

Reg. fairness: We have not explored this question in this work but included a remark on it in the broader impact section.

**Rev.#4**:

Reg. comparison to other methods: We do compare our method to *mean-optimal* policy $\pi(\mathbf{z})$, which is a standard method considered in the literature. Moreover, we have also included an additional baseline as explained in the reply to Reviewer 1.

[Meta-Review · NeurIPS 2020]

The reviewers and myself agree that the paper provides a strong conceptual contribution in analyzing robust off policy learning. There is some criticism on the presentation and the experimental part: 1) the authors are strongly encouraged to include the new stronger baseline that is described in their rebuttal, 2) the authors are strongly encouraged to discuss the benefits of their algorithm in settings with poor overlap. Despite these drawbacks the conceptual contribution of the work seems strong enough to merit acceptance.